# Behavior of Low-Cost Receivers in Base-Rover Configuration with Geodetic-Grade Antennas

**DOI:** 10.3390/s22072779

**Published:** 2022-04-05

**Authors:** Giannina Sanna, Tonino Pisanu, Salvatore Garau

**Affiliations:** 1Department of Civil, Environmental Engineering and Architecture, University of Cagliari, 09123 Cagliari, Italy; 2National Institute for Astrophysics, Astronomical Observatory of Cagliari, 09047 Selargius, Italy; tonino.pisanu@inaf.it; 3Poema S.r.l., 09122 Cagliari, Italy; salvatore.garau@poemaonline.eu

**Keywords:** low-cost GNSS receiver, u-blox ZED-F9P2, network RTK, land-surveying

## Abstract

The main goal of this research was to evaluate the performances of the ZED-F9P-Ublox low-cost GNSS receiver in a base-rover real configuration. We realized a base configuration with two permanent stations based on the ZED-F9P and two geodetic antennas and the rover configuration based on another ZED-F9P and an ANN-MB-00-00 Multi-band (L1, L2/E5b/B2I) active GNSS u-blox antenna. In the calculation of the reference stations, we compared the solutions with the ZED-F9P receiver and a professional receiver. Comparison showed greater variability in the solutions, but the coordinate values were in very good agreement. Standard deviations were in the order of a few millimeters. On the rover side, two car tests were performed in two different environments, one in an extra-urban environment with a long baseline of approximately 30 km in an open sky area with varying visibility and shielded locations, the other one in an urban area around a circle approximately 10 km in diameter with the presence of buildings and open sectors. The results of the measurements were very good, with more than 95% of fixed solutions in real-time and a time to fix on reacquisition of 1 or 2 s. Moreover, real-time kinematic solutions were in good agreement with the post-processed ones, showing that less than 5% of differences were above 30 mm in the horizontal component and 100 mm in the vertical component.

## 1. Introduction

Over the past few years, global satellite navigation systems have undergone considerable development, both in the space segment and the user segment [1]. For the completion of the BeiDou-3 positioning system, no fewer than 30 satellites were put into orbit within 2.5 years between 2017 and 2020 [2]. To date, 110 satellites are operational in the four navigation systems, of which 99 are currently operational in MEO orbit for worldwide coverage: GPS (29), GLONASS (24), GALILEO (22) and BeiDou (24). The possibility to obtain the satellite positioning using dozens of visible satellites has, therefore, become a reality very quickly. On the user segment side, several classes of users have been created, which need to be distinguished as follows: high volume devices for the consumer market, safety and liability critical devices, high precision devices, timing devices. GNSS receivers are continually evolving to offer improved performances tailored to each class but, as reported in [3], there is a trend towards using high-volume devices for professional applications. Indeed, dual frequency has not only become a strategic choice for high-end devices but is also entering the mid-range smartphone market.

To the best of our knowledge, at the time of writing, there are only three OEM receivers on the low-cost market, multi-frequency and multi-constellation: Septentrio SimpleRTK3B Pro (about 550 €), Bynav C1-8S (about 225 €) and u-blox ZED-F9P (approximately 175 €). All three can be used for absolute and relative RTK (Real-Time Kinematic) and NRTK (Network Real-Time Kinematic) positioning and may replace high-precision devices in typical land-surveying or geodetic applications, e.g., cadastral surveying, establishment of control points or even in research applications. Studies have been carried out during the last two years to evaluate the accuracy of the u-blox ZED-F9P, performing tests, usually devoted to hi-grade GNSS receivers and infrastructures [4,5,6], in order to assess their limitations.

Tests were addressed to investigate the performances of the low-cost dual-frequency for the estimation of atmospheric parameters. Krietemeyer et al. in [7], investigated the potential of the u-blox ZED-F9P in combination with a range of different quality antennas for the estimation of Zenith Tropospheric Delays (ZTDs). They showed that the receiver itself is very capable of achieving high-quality ZTD estimations.

Okoh et al. in [8] and Dan et al. in [9], found that the u-blox is a good candidate for TEC (Total Electron Content) studies, just like the high-cost receivers.

Some papers investigated ZED-F9P for landslide monitoring [10,11,12] and displacement detection. Hamza et al. [13] showed that on a basis of 30-min sessions the instruments can detect displacements from 10 mm upwards with a high level of reliability. In [14], Tunini et al. matched, in parallel, the ZED-F9P receiver with two high-price geodetic instruments, all connected to the same geodetic antenna. They processed the data, together with the observations coming from a network of GNSS permanent stations operating in that region. The results show that mm-order precision can be achieved by cost-effective GNSS receivers, and the results in terms of time series are largely comparable to those obtained using high-price geodetic receivers.

Some other tests were carried out to evaluate the performances of ZED-F9P, connected to a u-blox ANN-MB-00-00, as a land-survey receiver. Wielgocka et al. [15] investigated the performance of a u-blox ZED-F9P receiver, connected to a u-blox ANN-MB-00-00 antenna, during multiple field experiments. In static tests, the centimeter-level accuracy was achieved both for the horizontal and vertical components. In kinematic tests, the RMSE for the vertical component was 56.7 mm. For 46% of epochs, the positioning error for the height determination exceeded 5 cm, including 11% of epochs with errors exceeding 10 cm. In [16], with the same configuration, thirty-six monuments in four piedmont counties of South Carolina were surveyed in RTK FIX mode. The 95% confidence level accuracies for the horizontal and height dimensions were 3.7-cm and 4.2-cm (95%), respectively. Hamza et al. [17] compared the positioning quality from different types of low-cost antennas and analyzed the positioning differences between low-cost and geodetic instruments, demonstrating the excellent performance of the receiver, sufficiently appropriate for various geodetic applications. Krietemeyer et al. [18] presented an antenna field calibration method to improve the performance of the low-cost ANN-MB-00-00. After calibration, the ambiguity-fixed phase residuals maintain a Median Absolute Deviation (MAD) of 3.5 mm on L1 and 4.4 mm on L2, bringing its behaviour closer to that of geodetic antennas. Moreover, Janos and Kuras [19] evaluated the position accuracy using the u-blox ZED-F9P receiver, with the satellite signal supplied by both the dedicated low-cost u-blox ANN-MB-00-00 patch antenna and the Leica AS10 high-precision geodetic one. As a result, it was concluded that the ZED-F9P receiver equipped with a patch antenna is only suitable for precision measurements in conditions of wide-open sky areas. However, the configuration of this receiver with a geodetic-grade antenna significantly improves the quality of the results. In [20], Broekman and Gräbe demonstrated that a low-cost, mobile, real-time kinematic (RTK) geolocation service, fabricated with components readily available from commercial suppliers, provides centimeter-accuracy performance up to a distance of 15 km from the base station.

Starting from these results, the present study aims to assess the possibility that these receivers can work as spare units or even upgrade a real infrastructure of a network of permanent GPS stations to make it a multi-constellation network. In this case, the u-blox base or master [21,22] station would take advantage of the geodetic-grade antennas already present in the reference stations of the network. It is clear that it is necessary not only for the availability of the hardware but also of a very complete and well-organized software package that allows us to configure and evaluate all the possible performances of the receiver.

The performances can be evaluated from many different characteristics, for example, the possibility to operate as a base, the ability to power up the antenna and the ability to generate and process the RTK mode with the RTCM correction data. On the rover side, the Time To First Fix (TTFF) with a given number of satellites in view, the ability to set every possible configuration, for example the parameters of the antennas and so on.

In order to test the u-blox ZED-F9P receiver, we recreated the base-rover configuration similar to the one present in the network to be updated. The experimentation took place in the south of Sardinia, in an area covered by a network of NRTK (SARNET) permanent stations used for more than 15 years, in geodetic and topographic applications, both scientifically [23] concerning atmospheric parameter estimation and professionally for real-time precise positioning. The network is a GPS only infrastructure and needs to be expanded and upgraded. Two new reference stations were established, and two car tests were carried out in two different routes along suburban roads and in a city environment.

We created a small GNSS network and carried out tests according to the flow chart in Figure 1. First, we studied the ZED-F9P receiver to set the parameters and make it work as a base or as a rover in real-time. We set up two permanent stations, with geodetic-grade-antennas, and performed static surveys to determine their coordinates. During the static surveys we alternated the u-blox receiver with a professional one to compare the results. To evaluate the characteristics of the receiver as a rover, we performed two kinematic tests and compared the results obtained in real-time with those post-processed with the RTKLib software, widely tested and used by the scientific community. In the following, Section 2 describes the ZED-F9P receiver used to create the network infrastructure and Section 3 describes methods used to obtain the reference stations coordinates, the car test tracks and the quality assessment. In Section 4, the results and discussion of the experimentation are described in detail, while Section 5 reports the conclusions.

## 2. Materials

### 2.1. ZED-F9P Receiver

In our test, we used the SparkFun GPS-RTK2 Board, which houses the u-blox ZED-F9P [24] low-cost chip that can receive satellite signals in the lower and upper L-band (L1C/A, L1OF, E1, B1L, L2C, L2OF, E5b, B2l) from all available satellite constellations (GPS, GLONASS, Galileo, BeiDou, QZSS). The manufacturer states a positioning accuracy of 1 cm + 1 ppm in RTK mode, with a baseline limited to 20 km [25].

It offers RTK and RTN (Real-Time Network) operation with a high frequency measurement rate (up to 20 Hz) and accuracy (1 cm + 1 ppm). In conditions of good satellite visibility, the receiver quickly resolves its position (cold start < 24 s, reacquisition < 2 s).

Additionally, anti-jamming and anti-spoofing algorithms are implemented into the receiver, allowing the assumption that it can discard unwanted signals. It has a wide operating temperature range, low power consumption, light weight and a large number of physical inputs/outputs and communication capabilities. In the rover configuration, the receiver was connected to a standard u-blox ANN-MB-00-00 active patch antenna, having a circular ground plane (as recommended by the manufacturer). It was a Right-Hand Circular Polarized (RHCP) dual-band antenna (L1 and L2/E5b/B2I).

The u-blox has developed the u-center GNSS evaluation software for Windows [26], which allow to set and acquire many configurations, parameters and raw data to evaluate the receiver performances in real-time or to be used in post processing analysis. U-center has a clear and complete graphical user interface (Figure 2) where you can easily set and visualize many different parameters. The ZED-F9P can be set up with a great number of variables, such as those for the management of NMEA messages, those that make the receiver operate as a base or as a rover and those for the management of the RTCM3 protocol.

The ZED-F9P receiver can be set up to operate as a base/rover through the TMODE parameter. When it operates as a rover, the TMODE = 0 (disabled), to operate as a base, you set TMODE = 1 (survey-in), which means that the base must first estimate the antenna reference coordinates. You set TMODE = 2 (fixed) if the coordinates of the reference station are already known. When the receiver operates as a base, it only estimates the clock offset from the GPS time. We set the TMODE variable to the value of 2 because we already estimated the coordinates of the permanent stations.

Regarding the RTCM3, when operating as a rover, the receiver interprets the RTCM3 messages from 1001 to 4072, including 1007 antenna descriptor message. Operating as a base, however, the messages that the receiver can generate are far fewer and do not include antenna descriptors (Table 1). The antenna descriptors include two types of data: the PCO (phase center offset with respect to the ARP) on both L1 and L2, and the PCV (i.e., the phase center variations with respect to the signal arrival direction, mapped in elevation or azimuth and elevation). The ZED-F9P does not correct for PCO, but rover coordinates can be still corrected by referring the base antenna coordinates to the phase center and not to the ARP. The PCV correction instead, enters directly into the Double Differences equations [27] when using different antennas between base and rover and, therefore, cannot be corrected. The PCV values are usually very small: Table 2 shows the PCV values for the u-blox ANN-MB-00-00 antenna, always less than 5 mm for L1 and always less than 10 mm for L2 (Table 2).

In real-time activity, a server and a client of the Ntrip protocol for the transmission of the RTCM messages via the Internet are implemented in the u-center software. For post-processing activity, the software allows the creation of a log file, which contains all the information related to the configuration of the receiver and the raw data measurements. Afterwards, it is possible to obtain a RINEX file of the observations to post-process the phase and code observations with the commonly used GNSS processing software. However, all settings transmitted to the receiver can be sent outside the u-center software with simple scripts.

### 2.2. Reference Stations Setup

The only two permanent stations of the EPN network in Sardinia are located within the same area, housed on the terrace above our research laboratory, in Cagliari. The first one, UCAG00ITA (UCAG), is operated by ASI (Italian Space Agency) and has been included in the EPN network since August 2015. The second one, CAG100ITA (CAG1), is operated by our research group at the DICAAR of the University of Cagliari and has been included in the EPN network since January 2016. The two stations, which are separated by a few meters, are classified as a class A station, which means that can be used as a fiducial station for the EUREF densification.

We set up two new reference stations, estimating their position with a static survey from the permanent stations UCAG and CAG1. The first one, named POEM, was based in a lowland area approximately 12 km away from the EPN stations, at the “Sardegna Ricerche” research center site. POEM was equipped with a geodetic choke ring antenna ASH701945C_M from Ashtech, see Figure 3a. This antenna requires a 5 V power supply, which the ZED-F9P cannot provide. Therefore, we connected a Topcon GRS-1 survey-grade receiver in parallel, which provided the power supply. An antenna splitter, PD2120, was inserted together with a DC-Block to prevent the ZED-F9P from being powered by the voltage supplied from the Topcon receiver. Figure 3b shows the connections made between the antenna and the receivers. The second receiver allowed us to carry out independent data processing to determine the POEM coordinates.

The second reference station, named UFF1, was set up roughly twelve meters away from the CAG1 EPN station. UFF1 was equipped with a Trimble’s Zephir Geodetic 2 antenna [28], the same as those present in the network of GPS permanent stations. The antenna supports all bands (L1/L2/L5/G1/G2/G3/E1/E5ab/E6/B1/B2/B3) to which the receiver under investigation is connected. In this case, the ZED-F9P receiver was able to power up the antenna, which required a voltage of at least 3.5 V. The ZED-F9P receiver, in turn, was powered by the PC where the u-center resides. Figure 4 shows the position of the UFF1 reference station on the terrace of the building (a) and in relation to the EPN reference stations (b).

## 3. Methodology

### 3.1. Reference Stations Data Processing

At both stations, we first carried out observations to determine their coordinates, and then we made the receiver work as a base during kinematic tests. Observations to determine the position of the POEM reference station were recorded in 13 days, starting from DOY 64 to DOY 173, between March and June 2021, as summarized in Table 3. Observations were made for more than 200 h. Approximately 85 h were observed using the Topcon GRS1 receiver and approximately 115 h with the u-blox ZED-F9P receiver. For station UFF1, observations were made during 4 days for a total of more than 79 h (Table 4).

The output log files of the master and rover ZED-F9P were converted to RINEX 3.1 standard files with the RTKLib convbin.exe utility. The observations were then post-processed with the demo5 b34b version of the RTKLib software [29]. The antenna phase center corrections were applied to all the stations involved using values from the IGS product igs14.atx. To determine the coordinate of the POEM and UFF1 base stations, we performed the calculation of the baselines UCAG-POEM, CAG1-POEM and the baselines UCAG-UFF1 and CAG1-UFF1 in daily sessions. Then, starting from the daily estimates, we determined the coordinate averages and corresponding standard deviations relative to each baseline. Lastly, we estimated the final coordinates and accuracy as mean values from both baselines.

POEM coordinate computation was performed according to the option summarized in Table 5 for the long baseline. The values were estimated from UCAG and CAG1 with the Topcon-GRS1 receiver, and likewise with the ZED-F9P receiver.

UFF1 post-processing was performed according to the options listed in Table 5 for the short baseline. Only the L1 frequency was selected given the very short baseline involved.

### 3.2. Car Tests

Two car tests were carried out: “car-test 1” and “car-test 2”, characterized by different environmental/geographic conditions. The car-test 1 route was carried out on an extra-urban road, characterized by variable orographic conditions (from flat to mountainous) (Figure 5). The car departed from the Sardegna Ricerche base, where the reference station POEM was housed, and drove westwards for approximately 30 km. At the furthest point from the permanent station, we crossed the small town of Villamassargia (CA). After stopping for a few minutes, we resumed our journey to the starting point in POEM. A total of approximately 64 km were covered.

The car-test 2 was conducted on an urban route with a variable track that passed from streets bordered by tall buildings to free streets. The route was approximately 25 km long and deviated from the reference station for approximately 10 km (Figure 6). The survey started near the permanent station UFF1 and ended a few kilometers away from UFF1, after running approximately 29.4 km. In both tests, the base stations sent the RTCM messages listed in Table 2, by the Ntrip Server integrated in u-center, with a frequency of 1 Hz and were connected to the internet with a static IP number. During the tests, the base station also recorded the raw measurements for subsequent post-processing.

The rover receiver was placed on the car roof firmly locked with a magnetic mount and was provided with a metal plate at the base of the u-blox antenna (Figure 7). The antenna cable arrived inside the car where the ZED-F9P was housed and connected via the USB port to a Win10 pc/tablet, where the u-center software was running. The rover was set to acquire satellites with a cutoff of 10°. A wireless connection to a popular telephonic company provided the internet link. Using the Ntrip client integrated in the u-center software, we connected it to the Ntrip server.

Similar to the base station, the rover during testing recorded a log file. The information recorded in the log file consisted of: NMEA messages (GNS, GRS, GSA, GST, GSV), PUBX messages (u-blox NMEA extension) that summarize the real-time positioning results, and proprietary RAWX and SFRBX messages containing raw observations and navigation data.

Kinematic surveys were post-processed with RTKLib, according to the options summarized in Table 5.

### 3.3. Quality Assessment

Both real-time and post-processed solutions were provided as global geodetic Earth-Centered Earth-Fixed (ECEF) coordinates but, in order to compare coordinates in terms of horizontal and vertical components, we transformed those differences to a local-level reference system of North, East, Up (NEU). The Up axis coincides with the ellipsoidal normal in the point, while East and North axes belong to the tangent plane in the same point.

Denoting with $X_{i}^{PP}$
*=* (*x_i_*, *y_i_*, *z_i_*)*^PP^* and $X_{i}^{RT}$
*=* (*x_i_*, *y_i_*, *z_i_*)*^RT^*, the vectors corresponding to the post-processed and real-time ECEF coordinates at the same epoch *i*, we evaluated the vector $X_{i}^{PR}$
*=*
$X_{i}^{RT}$ − $X_{i}^{PP}$.

Applying the global to local-level system transformation (1) to the vector $X_{i}^{PR}$, we obtained the NEU component of the difference between real-time and post-processed coordinates. Indicating $x_{i}^{PR}$ with the NEU vector, we have:(1)$$x_{i}^{PR}=R^{T}X_{i}^{PR}$$
being:(2)$$R=\left(\begin{array}{ccc} -sin\varphi_{i}^{PP}\;cos\lambda_{i}^{PP} & -sin\lambda_{i}^{PP} & cos\varphi_{i}^{PP}\;cos\lambda_{i}^{PP}\\ -sin\varphi_{i}^{PP}\;sin\lambda_{i}^{PP} & cos\lambda_{i}^{PP} & cos\varphi_{i}^{PP}\;sin\lambda_{i}^{PP}\\ cos\varphi_{i}^{PP} & 0 & sin\varphi_{i}^{PP} \end{array}\right)$$
(3)$$x_{i}^{PR}=\left(\begin{array}{c} n_{i}\\ e_{i}\\ u_{i} \end{array}\right)$$
and $\varphi_{i}^{PP},\;\lambda_{i}^{PP}$, the latitude and longitude of the post-processed point at epoch *i*.

Then we evaluated the mean values:$$\bar{n}=\frac{1}{N}\overset{N}{\underset{1}{\sum }}e_{i}\;;\;\;\bar{e}=\frac{1}{N}\overset{N}{\underset{1}{\sum }}e_{i}\;;\;\bar{u}=\frac{1}{N}\overset{N}{\underset{1}{\sum }}u_{i}$$
and the figures:(4)$$dH_{i}^{PR}=\sqrt{n_{i}^{2}+e_{i}^{2}}\;,dU_{i}^{PR}=\sqrt{u_{i}^{2}}$$

Moreover, during the car tests, the real-time position quality of the *i*-th point, as provided by the u-center, was split in a horizontal position precision (the term accuracy is instead used in the u-center software user guide) (PACC*_H_*)*^RT^* and a vertical position precision (PACC*_V_*)*^RT^*. Otherwise, in RTKLib, the quality of the solutions was given as standard deviation ${\left(\sigma_{x},\;\sigma_{y\;,}\sigma_{z}\right)}_{i}$ in global geodetic coordinates. Therefore, in order to compare the post-processed values with the RT values, the ECEF standard deviations were converted to local-level standard deviations with the following transformation:(5)$$\left(\begin{array}{c} \sigma_{n}\\ \sigma_{e}\\ \sigma_{u} \end{array}\right)=R^{T}\left(\begin{array}{c} \sigma_{x}\\ \sigma_{y}\\ \sigma_{z} \end{array}\right)$$

Then we put:(6)$${\left({\mathrm{PACC}}_{H}\right)}^{PP}=\pm \sqrt{\sigma_{n}{}^{2}+\sigma_{e}{}^{2}}\;\;\;\;\;\;\;\;{\left({\mathrm{PACC}}_{V}\right)}^{PP}=\pm \sigma_{u}$$
being (PACC*_H_*)*^PP^* and (PACC*_V_*)*^PP^*, the horizontal and vertical position precision of the point estimated in post-processing.

Finally, we estimated the RMSE both for the horizontal and vertical components, for real-time as:(7)$${\left({\mathrm{RMSE}}_{H}\right)}^{RT}=\sqrt{\frac{1}{N}\overset{N}{\underset{n=1}{\sum }}{\left|{\left({\mathrm{PACC}}_{H}\right)}^{RT}{}_{n}\right|}^{2}};\;\;\;\;\;\;\;{\left({\mathrm{RMSE}}_{V}\right)}^{RT}=\sqrt{\frac{1}{N}\overset{N}{\underset{n=1}{\sum }}{\left|{\left({\mathrm{PACC}}_{V}\right)}^{RT}{}_{n}\right|}^{2}}\;\;\;$$
and for post-processed solutions as:(8)$${\left({\mathrm{RMSE}}_{H}\right)}^{PP}=\sqrt{\frac{1}{N}\overset{N}{\underset{n=1}{\sum }}{\left|{\left({\mathrm{PACC}}_{H}\right)}^{PP}{}_{n}\right|}^{2}};\;\;\;\;\;\;\;\;{\left({\mathrm{RMSE}}_{V}\right)}^{PP}=\sqrt{\frac{1}{N}\overset{N}{\underset{n=1}{\sum }}{\left|{\left({\mathrm{PACC}}_{V}\right)}^{PP}{}_{n}\right|}^{2}}$$
being *N* the total number of points.

## 4. Results and Discussion

### 4.1. Reference Stations

During the implementation of the POEM permanent station, the ZED-F9P receiver worked continuously for several days. Table 6 lists the standard deviation of the averaged daily solutions in X, Y and Z and NEU showing that they are greater with the ZED-F9P receiver both from the UCAG and CAG1 EPN stations than those with the Topcon receiver. The calculation showed greater variability in the solutions obtained with the u-blox receiver. Indeed, we obtained standard deviations always lower than 1 cm horizontally, but of the order of 4 cm in altitude.

Table 7 shows the differences in POEM coordinates, both ECEF and local-level, separately for the Topcon and the u-blox receivers, obtained from UCAG and from CAG1. Albeit differences in standard deviations, mean values of coordinates were in good agreement and were, therefore, averaged altogether in order to obtain a mean solution from UCAG, a mean solution from CAG1 and, eventually, the final estimate (Table 8).

In UFF1, similar results were obtained. Table 9 shows the differences, both in terms of ECEF values and local-level coordinates, obtained from UCAG and from CAG1. A final solution was obtained averaging all values (Table 10).

For both the POEM and UFF1 reference stations surveys, we used the observations of both EPN stations, UCAG and CAG1. For POEM, the calculation revealed a vertical difference of −21.5 mm, with the Topcon receiver, and −37.2 mm with the ZED-F9P receiver. For UFF1, the calculation revealed a vertical difference of −25.9 mm with the ZED-F9P receiver. Although these values are still acceptable, the concordance between the POEM and UFF1 closure loop error, calls for an investigation into the height difference between UCAG and CAG1. In fact, although the two stations are only a few meters apart, they are not yet co-located, meaning that there is no local survey to determine their mutual position.

### 4.2. Car Tests

During car test 1, in real-time, we obtained 4559 fixed positions corresponding to 97.23%, with the number of satellites used in the solution varying between 21 and 32 (Figure 8a). The fixed solution was lost 7 times, and the average reacquisition time of the fixed position was 18 s. The (RMSE*_H_*)*^RT^* and (RMSE*_V_*)*^RT^* of the fixed positions, computed according to (7), is of 14.5 mm and 13.3 mm in the horizontal and vertical positions, respectively. Figure 9 describes the horizontal (a) and vertical (b) precision distribution of fixed positions along the track, obtained in real-time. The figure shows values between 14 and 50 mm in the horizontal position and between 10 and 86 mm in elevation.

Post-processing, performed according to the settings in Table 5 for the kinematic survey, produced 90.61% of the fixed positions, with the number of satellites used in the solution ranging from 5 to 22 (Figure 8b). According to (5), (6) and (8), the (RMSE*_H_*)*^PP^* and (RMSE*_V_*)*^PP^* were 5.7 mm and 17.4 mm. Figure 10 describes the horizontal (a) and vertical (b) precision distribution of the fixed positions along the track, obtained in post-processing, while Table 11 summarizes and compares the values of real-time and post-processed solutions. The percentage of fixed positions was 7% higher for the u-blox than the RTKLib solution. This figure correlated with the number of satellites used in the solution, which in RTKLib was always lower than in the ZED-F9P. In contrast, the planimetric precision achieved by RTKLib was significantly higher than that of ZED-F9P, while the elevation precision was equivalent.

Finally, according to (3) and (4), we compared the post-processed coordinates with the real-time ones, evaluating differences in terms of the horizontal and vertical components. Mean differences in *e*, *n* and *u* were 2.4 mm, 4.9 mm and −0.8 mm, respectively. Moreover, despite a few peaks, 95% of differences were below 30 mm in the horizontal component and 90 mm in the vertical component. Figure 11 shows horizontal (a) and vertical (b) differences between real-time and post-processed positions, related to the number of satellites (SVs) used in post-processed solutions.

Car test 2 started at 09:48:43 and ended at 10:53:57, for a total of 3915 epochs/seconds. In real-time, we obtained 3726 fixed positions corresponding to 95.17%, with the number of satellites used in the solution varying between 27 and 32 (Figure 12a). The fixed solution was lost 19 times, and the average reacquisition time of the fixed position was 8 s. The (RMSE*_H_*)*^RT^* and (RMSE*_V_*)*^RT^* of the fixed positions were 14.1 mm and 11 mm, respectively. Figure 13 describes the horizontal and vertical precision distribution of the fixed positions along the track, obtained in real-time. Values span between 14 and 38 mm in the horizontal position and between 10 and 57 mm in elevation.

Post-processing produced 88.8% of the fixed positions with satellites ranging from 5 to 29 to get the solution (Figure 12b). (RMSE*_H_*)*^PP^* and (RMSE*_V_*)*^PP^* were 4 mm and 10.7 mm, respectively. Figure 14 describes the horizontal and vertical precision distribution of fixed positions along the track, obtained in post-processing, while Table 12 summarizes and compares values of real-time and post-processed solutions. Again, the percentage of fixed positions was more than 6% higher for the u-blox than that of the RTKLib solution, and the planimetric precision achieved by RTKLib was significantly higher than that of ZED-F9P.

As for car test 1, we compared the post-processed coordinates with the real-time ones. Mean differences in *e*, *n* and *u* were 0.9 mm, 2.4 mm and −26.8 mm, respectively. Moreover, 95% of differences were below 30 mm in the horizontal component and 100 mm in the vertical component. Figure 15 shows the horizontal (a) and vertical (b) differences between real-time and post-processed positions in relation to the number of satellites (SVs) used in post-processed solutions.

In both car tests, the differences between real-time and post-processed positions were limited to 3 cm in horizontal and 10 cm in altitude for 95% of the comparable epochs. Table 13 summarizes and compares the results for car test 1 and car test 2.

## 5. Conclusions


We set ourselves the goal of evaluating the behavior of the u-blox ZED-F9P receiver for use in geodetic and topographic applications and assessing its behavior both as a base and as a rover. In base mode, the RTCM messages enabled on ZED-F9P do not allow the sending of the full description of the antenna to the rover, so it is essential for the rover to know the model of the antenna connected to the base in advance. As an alternative, it is necessary to send in the RTCM 1005 message the coordinates referred to the phase center of the antenna and not to the ARP. It is not unlikely that a firmware upgrade will fix the problem. Regardless of the RTCM problem, the results on the base side will allow us to proceed with the experimentation and, thus, to place the ZED-F9P receiver within the network of permanent geodetic stations.

On the rover side, the situation is more complex. The ZED-F9P receiver, as already highlighted in [19], does not currently handle the PCV antenna calibration information. Hence, it will be difficult to go below the claimed accuracy in real-time applications without the constraint of using the same model of antenna for both the base and the rover. In our opinion, this limits the use of this receiver and antenna in deformation study applications or in tropospheric parameter estimation as underlined in [18]. RTKLib, widely used by the scientific community, could be used to solve this limit. RTKLib accepts streams of data from both the base and the rover and calculates the rover’s position in real-time, using its own RTK engine instead of the receiver engine. RTKLib is capable of correcting PCV, provided that the antennas are present in the antenna database.

The comparison between the real-time and the post-processed solutions allowed us to evaluate the efficiency of the RTK engine of the ZED-F9P, which was higher than that of RTKLib. The RTKLib solutions were given with a higher precision, but real-time solutions were in good agreement with the post-processed ones, showing that less than 5% of differences were above 30 mm in the horizontal component and 100 mm in the vertical component. In conclusion, comparing the RTKLib and the u-blox RTK engines, the u-blox RTK proved to be reliable and more efficient and, therefore, difficult to leave.

The study carried out so far is the starting point for performing other tests to answer the many questions still pending. For example, the comparison with professional rover receivers, the use of a survey-grade antenna on the rover side and the evaluation of the receiver range both in terms of accuracy and reliability at long distances. In the future, the tests indicated above, as well as a more complete analysis of the daily solutions of the permanent stations, will allow us to more fully exploit the potential of the receivers of this low-cost market segment.

## Figures and Tables

**Figure 1 sensors-22-02779-f001:**
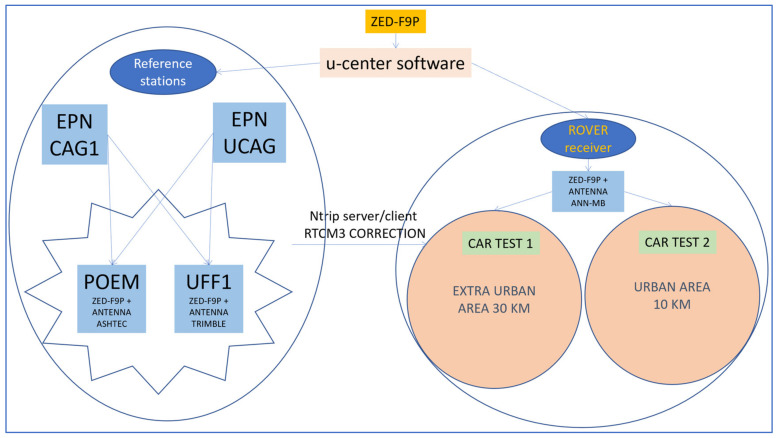
Flowchart of the activity.

**Figure 2 sensors-22-02779-f002:**
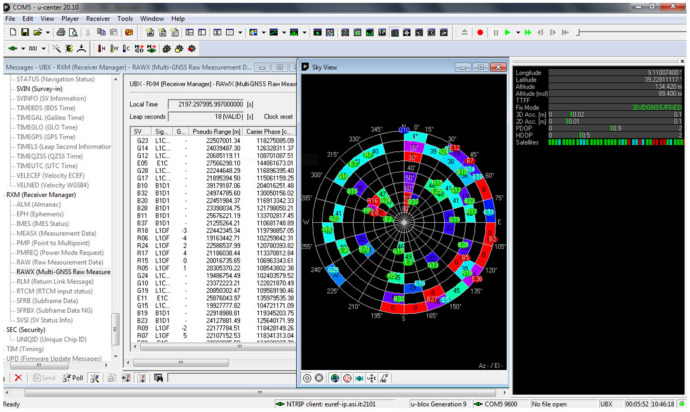
U-center ver. 20.10 user interface.

**Figure 3 sensors-22-02779-f003:**
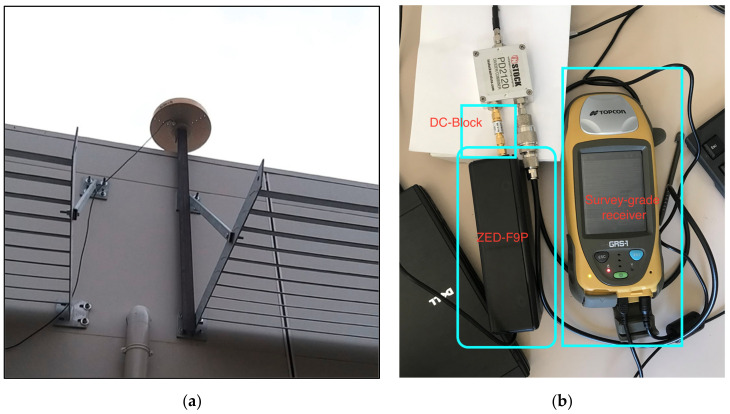
POEM reference station: (**a**) antenna setting, (**b**) antenna cable connection to the receivers.

**Figure 4 sensors-22-02779-f004:**
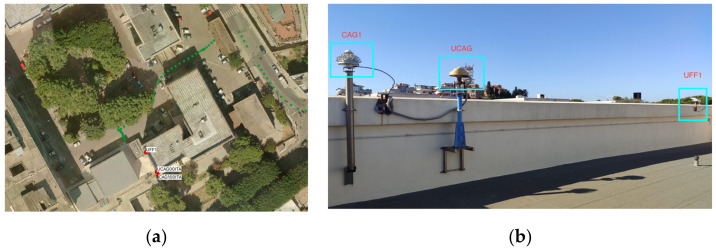
UFF1 reference station: (**a**) image of the area where the antenna is placed, (**b**) position of UFF1 with respect to EPN stations.

**Figure 5 sensors-22-02779-f005:**
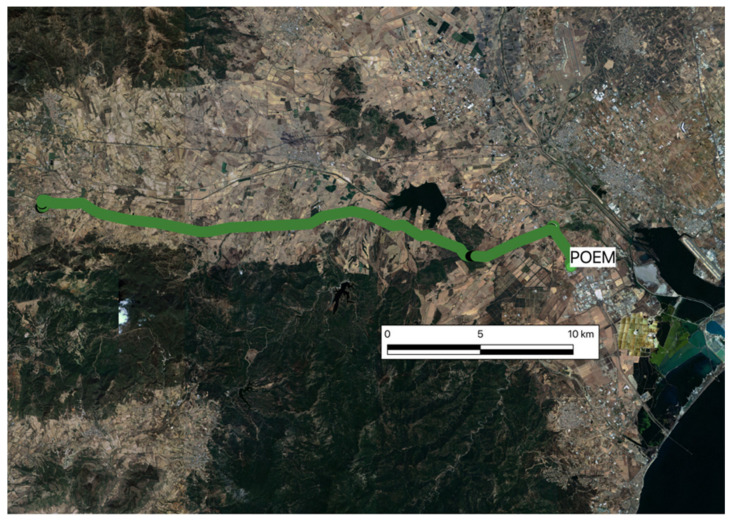
Area and track of the car test 1round-trip.

**Figure 6 sensors-22-02779-f006:**
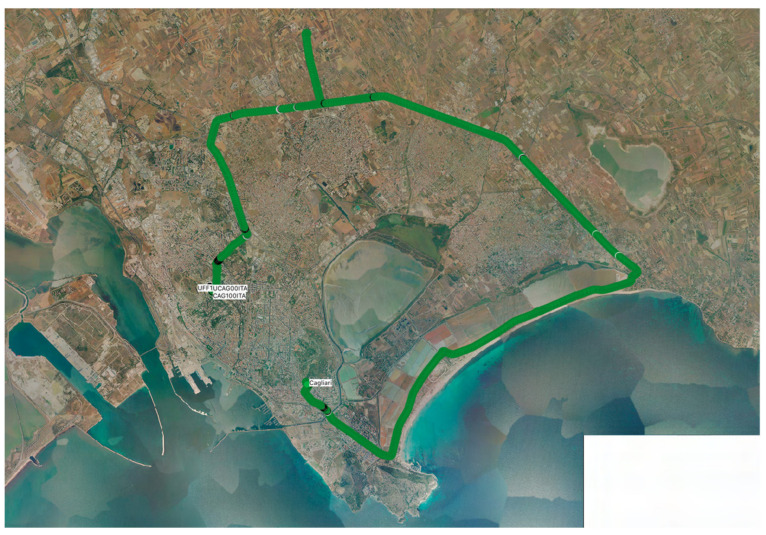
Area and route of car test 2.

**Figure 7 sensors-22-02779-f007:**
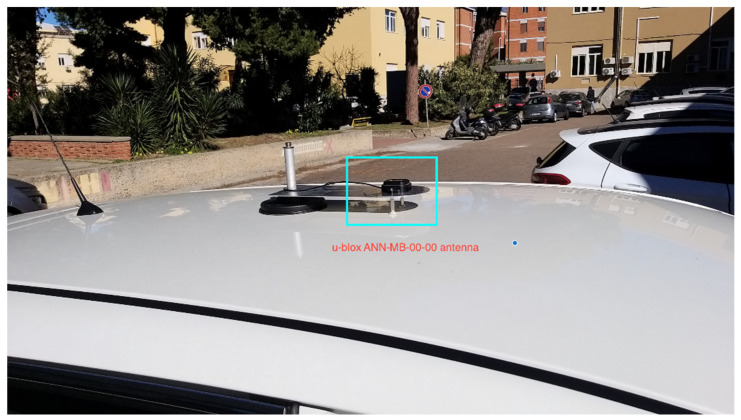
U-blox antenna setting on the car roof.

**Figure 8 sensors-22-02779-f008:**
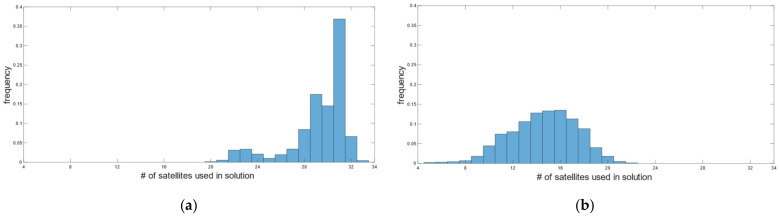
Car test 1: distribution of the number of satellites used in real-time positioning (**a**) and in the post-processed RTKLib solutions (**b**).

**Figure 9 sensors-22-02779-f009:**
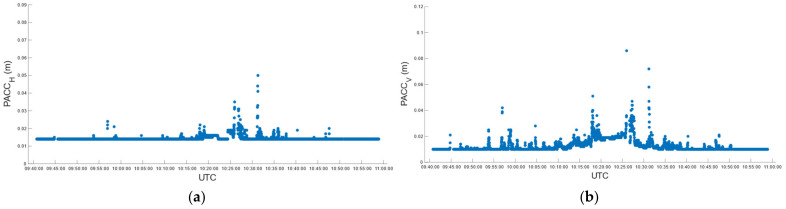
Car test 1: real-time estimated precision, (PACC*_H_*)*^RT^* (**a**) and (PACC*_V_*)*^RT^* (**b**), over time.

**Figure 10 sensors-22-02779-f010:**
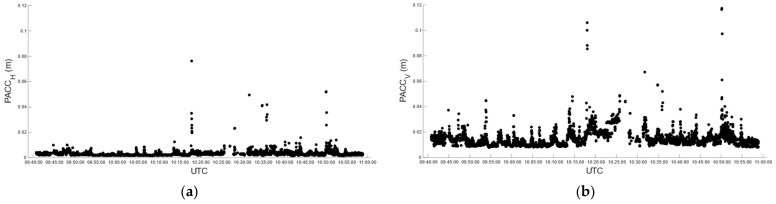
Car test 1: post-processed estimated precision, (PACC*_H_*)*^PP^* (**a**) and (PACC*_V_*)*^PP^* (**b**), over time.

**Figure 11 sensors-22-02779-f011:**
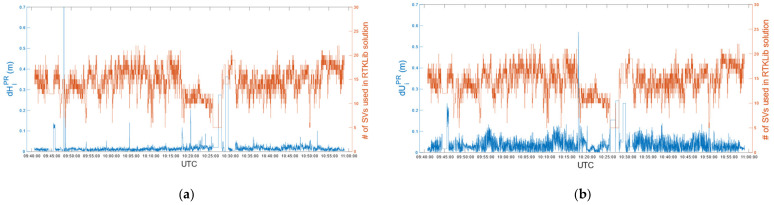
Car test 1: Horizontal (**a**) and vertical (**b**) differences between real-time and post-processed positions related to the # of satellites (SVs) used in post-processed solutions, epoch by epoch.

**Figure 12 sensors-22-02779-f012:**
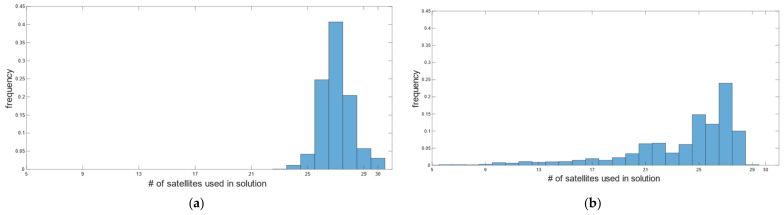
Car test 2: distribution of the number of satellites used in real-time positioning (**a**) and in the post-processed RTKLib solutions (**b**).

**Figure 13 sensors-22-02779-f013:**
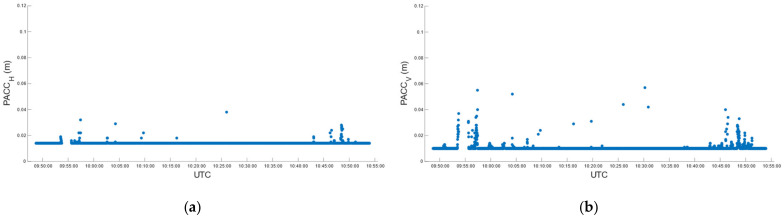
Car test 2: real-time estimated precision, (PACC*_H_*)*^RT^* (**a**) and (PACC*_V_*)*^RT^* (**b**), over time.

**Figure 14 sensors-22-02779-f014:**
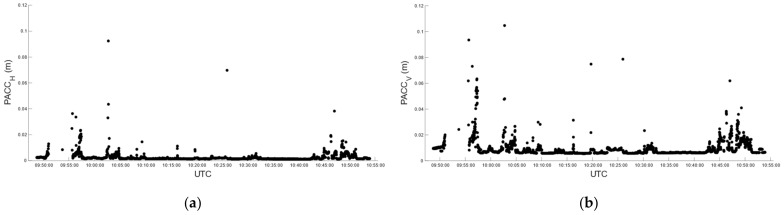
Car test 2: post-processed estimated precision, (PACCH)*^PP^* (**a**) and (PACCV)*^PP^* (**b**), over time.

**Figure 15 sensors-22-02779-f015:**
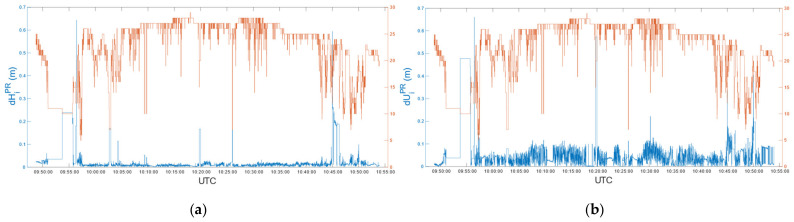
Car test 2: Horizontal (**a**) and vertical (**b**) differences between real-time and post-processed positions related to the # of satellites (SVs) used in post-processed solutions, epoch by epoch.

**Table 1 sensors-22-02779-t001:** List of broadcasted RTCM output messages.

Message Type	Description
RTCM 1005	Stationary RTK reference station ARP
RTCM 1077	GPS MSM7
RTCM 1087	GLONASS MSM7
RTCM 1097	Galileo MSM7
RTCM 1127	BeiDou MSM7
RTCM 1230	GLONASS code-phase biases

**Table 2 sensors-22-02779-t002:** U-blox ANN-MB-00-00 antenna phase center properties.

GNSS	Phase Center Offset (PCO)		Phase Center Variation (PCV)
	Horizontal plane	Up	Overall azimuth/elevation
GPS L1 C/A	<5 mm in all directions	8.9 mm	<5 mm
GPS L2C	<5 mm in all directions	7.6 mm	<10 mm

**Table 3 sensors-22-02779-t003:** List of survey days, duration and receiver used at POEM reference station.

DOY	Start Time	Stop Time	Span	Receiver
64	“12:26:00.0”	“18:21:30.0”	05:55:30	UBLOX-F9P
67	“09:15:00.0”	“19:56:00.0”	10:41:00	UBLOX-F9P
77	“17:47:30.0”	“23:59:30.0”	06:12:00	UBLOX-F9P
78	“00:00:30.0”	“18:14:00.0”	18:13:30	UBLOX-F9P
161	“10:34:30.0”	“23:59:30.0”	13:25:00	Topcon GRS1
162	“00:00:00.0”	“23:59:30.0”	23:59:30	Topcon GRS1
163	“00:00:00.0”	“23:59:30.0”	23:59:30	Topcon GRS1
164	“00:00:00.0”	“23:59:30.0”	23:59:30	Topcon GRS1
165	“16:58:30.0”	“23:59:30.0”	07:01:00	UBLOX-F9P
166	“00:00:00.0”	“23:59:30.0”	23:59:30	UBLOX-F9P
167	“00:00:00.0”	“23:59:30.0”	23:59:30	UBLOX-F9P
172	“10:42:30.0”	“14:10:00.0”	03:27:30	UBLOX-F9P
173	“00:00:00.0”	“16:22:00.0”	16:22:00	UBLOX-F9P

**Table 4 sensors-22-02779-t004:** List of survey days, duration and receiver used at UFF1 reference station.

DOY	Start Time	Stop Time	Span
34	“15:47:30.000”	“23:59:30.0”	08:12:00
35	“00:00:00.000”	“08:18:30.0”	08:18:30
39	“10:36:30.000”	“23:59:30.0”	13:23:00
40	“00:00:00.000”	“09:47:00.0”	09:47:00

**Table 5 sensors-22-02779-t005:** List of different options set in the demo5-b34b RTKLib software for the long and short baseline and kinematic survey.

Option	Long Baseline	Short Baseline	Kinematic Survey
pos mode	Static	Static	Kinematic
freqs	L1+L2/E5b	L1	L1+L2/E5b
solution	Combined-Phase Reset	Forward	Combined-Phase Reset
elev mask	5.0 deg	5.0 deg	10 deg
ionos opt	Estimate TEC	OFF	Broadcast
tropo opt	Saastamoinen	OFF	Saastamoinen
ephemeris	Broadcast	Broadcast	Broadcast
navi sys	GPS GLONASS Galileo BDS SBAS	PS GLONASS Galileo BDS	GPS GLONASS Galileo BDS
amb res	Continuous	Continuous	Fix and Hold
Rec. antenna PCV correction file	igs14.atx	igs14.atx	igs14.atx

**Table 6 sensors-22-02779-t006:** POEM ECEF coordinate standard deviations.

Receiver	from	Std x (mm)	Std y (mm)	Std z (mm)	Std e (mm)	Std n (mm)	Std u (mm)
Topcon GRS1	UCAG	9.7	4.2	6.8	2.6	1.2	12.3
Topcon GRS1	CAG1	9.3	5.1	6.1	3.6	1.6	11.6
UBLOX-F9P	UCAG	32.1	13.6	24.2	8.5	2.7	41.5
UBLOX-F9P	CAG1	26.9	8.7	24.3	4.4	1.1	37.0

**Table 7 sensors-22-02779-t007:** Differences in POEM coordinates obtained from UCAG and CAG1.

		dx (mm)	dy (mm)	dz (mm)	de (mm)	dn (mm)	du (mm)
Topcon GRS1	CAG1-UCAG	−18.6	−1.6	−11.2	1.3	3.1	−21.5
UBLOX-F9P	CAG1-UCAG	−26.3	−5.6	−25.8	−1.4	−3.0	−37.2

**Table 8 sensors-22-02779-t008:** Final ECEF coordinates and accuracy, expressed as ECEF and NEU, of POEM station.

	x (m)	y (m)	z (m)	σ_x_ (mm)	σ_y_ (mm)	σ_z_ (mm)	σ_e_ (mm)	σ_n_ (mm)	σ_u_ (mm)
POEM	4,885,504.830	771,308.545	4,013,699.840	16.9	3.1	15.1	0.4	0.8	22.9

**Table 9 sensors-22-02779-t009:** Differences in UFF1 coordinates obtained from UCAG and CAG1.

	dx (mm)	dy (mm)	dz (mm)	de (mm)	dn (mm)	du (mm)
CAG1-UCAG	−22.0	−2.6	−13.9	0.9	3.2	−25.9

**Table 10 sensors-22-02779-t010:** Final coordinates and accuracy of UFF1 station.

	x (m)	y (m)	z (m)	σ_x_ (mm)	σ_y_ (mm)	σ_z_ (mm)	σ_e_ (mm)	σ_n_ (mm)	σ_u_ (mm)
UFF1	4,885,047.979	783,338.893	4,012,051.043	15.6	1.8	9.8	0.6	2.3	18.3

**Table 11 sensors-22-02779-t011:** Car test 1: summary of figure values.

	% of Fixed Positions	Min–Max Sat Used	RMSE*_H_* (mm)	RMSE*_V_* (mm)
Real-Time	97.23	21-32	14.5	13.3
Post-Processed	91.43	5-22	5.7	17.4

**Table 12 sensors-22-02779-t012:** Car test 2: summary of figure values.

	% of Fixed Positions	Min–Max Sat Used	RMSE*_H_* (mm)	RMSE*_V_* (mm)
Real-Time	95.17	27-32	14.1	11
Post-Processed	88.8	5-29	4	10.7

**Table 13 sensors-22-02779-t013:** Summary of car test 1 and car test 2 results.

	% of Fixed Position		RMSE*_H_* (mm)		RMSE*_V_* (mm)		$\bar{e}$	$\bar{n}$	$\bar{u}$
	RT	PP	RT	PP	RT	PP	(mm)	(mm)	(mm)
Car test 1	97.23	90.61	14.5	5.7	13.3	17.4	2.4	4.9	−0.8
Car test 2	95.17	88.8	14.1	4	11	10.7	0.9	2.4	−26.8

## Data Availability

The source data can be sent after an e-mail inquiry.

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
