# Peer review of "Behavior of Low-Cost Receivers in Base-Rover Configuration with Geodetic-Grade Antennas"

_sensors, 2022, doi:10.3390/s22072779_

Round 1

Reviewer 1 Report

See attached

Author Response

Overall Comments

The paper evaluates U-blox ZED-F9P for high precision positioning. The significance of this paper is very low since this device has been evaluated elsewhere and found suitable for high precision application and is integrated in many applications such as autonomous driving and robotics. See this article for example:

Michael E. Hodgson (2020) On the accuracy of low-cost dual-frequency GNSS network receivers and reference data, GIScience & Remote Sensing, 57:7, 907-923, DOI: 10.1080/15481603.2020.1822588.

Response: We would like to thank the reviewer. for his suggestion and for the comment that forced us to think better about the contents of our work.  We have read with much interest the paper indicated by him that, together with other works we have analyzed, performs tests to evaluate the accuracy of the instrument under discussion. In several of the papers analyzed, the ZED-F9P receiver is tested as a rover, using professional permanent stations, or networks of permanent stations, as reference stations. Only one work (Broekman, A.; Gräbe, P.J. A Low-Cost, Mobile Real-Time Kinematic Geolocation Service for Engineering and Research Applications. HardwareX 2021, 10, doi:10.1016/j.ohx.2021.e00203.) studies a configuration in which both the base station and the rover are equipped with the u-blox receiver to build an entire infrastructure based on it. Our work is part of this line of research, trying to expand, diversifying, the experiments carried out so far. In order to make our contribution to the research clearer, a sentence has been added in the introduction.

The methodology for assessment of real-time positioning accuracy does not give a true reflection of the accuracy. The solutions from the same data processed in real-time and post-processed is compared. The better option would have been to have another geodetic receiver in the car connected to the same antenna by a splitter with DC block on one end (just like POEM setup) to compare the solutions of the F9P to.

Response: We completely agree with the referee.The term accuracy was used in the u-center manual without any reference and from the way it is produced in the sotware, it is more properly a precision.  In order to improve the understanding of the text we have corrected the term and indicated in the note the word used by the manufacturer. Surely a second receiver connected to the same antenna as the ZED-F9P and with corrections coming from an independent permanent station would have given us a comparison solution. In this experimentation the purpose was mainly to evaluate the possibility to introduce the receiver in a network of permanent stations and to assess the reliability and continuity of operation.

The paper claims this receiver may be used for surveying. For this I am convinced about the ZED-F9P receiver but have doubts about the ANN-MB-00 antenna suitability for surveying. How can the antenna be mounted directly above a survey mark? For heighting purposes the ARP of the antenna must be measured above ground height and the phase centre offset from ARP must be known. Also the phase centre must be plum with the survey mark by having the antenna screwed on top of a survey tribrach. Does the ANN-MB-00 meet these requirements.

Response: We completely agree with the referee and in fact have reached the same conclusion as indicated in lines 441 and 442. We indicated only the receiver but meant to refer to both the receiver and the antenna. The term has been added in the new version.

Other Comments

Pg 1 L32: Should also mention QZSS which F9P can also track

Response: The QZSS positioning system can also be tracked by the ZED-F9P receiver, but it does not have worldwide coverage. Its coverage is limited to the Asia-Pacific region so the satellites of the constellation cannot be tracked in Europe. In the description we intended to refer only to satellites in MEO orbit with worldwide coverage. However ZED-F9P can track QZSS satellites, so we added the in formation in the receiver description. 

Pg 2 L88 Define what is “u-blox master station”?

Pg 3 L98 “master-rover configuration” – what does this mean? Usually the term base-rover or reference-rover is used for RTK positioning

Response: The term master is synonymous with base, it was already used in the book Schofield, W.; Breach, M. Engineering Surveying, Sixth Edition (2007) at page 338; more recently Dabove, P. (The Usability of GNSS Mass-Market Receivers for Cadastral Surveys Considering RTK and NRTK Techniques. Geod. Geodyn. 2019, 10, 282-289, doi:10.1016/J.GEOG.2019.04.006) explained the meaning of the term. Evidently this is not a commonly used term, so it was replaced with the base term as suggested by the reviewer. A reference has been added to the first sentence where it is used.

Pg 3 L114: define what is RTN? “high frequency” – does this refer to measurement rate?

Response: thanks for your careful reading, we have added the meaning of the RTN acronym and added "measurement rate" in the following sentence

Pg4 Table 2: At what rate was each of the RTCM messages broadcast to rover?

Pg4 L152: Was a NTRIP server setup in the base receiver? Was an NRTIP client setup for rover to receive corrections? Was transmission done through Wifi? Were the receivers given IP addresses?

Response: All the information and clarifications requested have been inserted in the text, rearranging it, in the lines from 229 to 232

Pg 4 L162 I don’t see the point of describing UCAG and CAG1 unless they are not used in data analysis. If so, must specify why these stations are described.

Response: UCAG and CAG1 were used to determine the coordinates of POEM and UFF1 stations in the ETRF2000 reference frame. A sentence was added in the lines 190-191 to clarify their use.

Pg5 L174: Should mention that a antenna spitter was used with DC block from ZED-F9P and power in from TOPCON.

Response: the information was added at line 196

Pg6 L219: How was the RTK positioning results logged? Was it NMEA, raw u-blox or RTCMv3 format?

Response: The information recorded in the log file consisted of NMEA messages (GNS, GRS, GSA, GST, GSV), PUBX (u-blox NMEA extension) messages that summarize the real time positioning results and proprietary RAWX and SFRBX messages containing the raw observations and navigation data.  The latter two types of data are what the convbin.exe software uses to build the rinex file. To make the description of the experiment clearer, we have added this information from line 228 to line 232 and 242-245.

Pg6 L205: GCAG why is this mentioned? Was it used to data analysis?

Response: GCAG was not subsequently used for analysis. Its reference in the text was deleted and the sentence rewritten.

P225 Section 223: It is not clear how you assessed the real-time and post processed position accuracy? Did you match the solutions at the same epoch and compute differences and RMS of differences? Or did you take the stds from RTKLib and u-center and use this to compute RMSE?

Response: We did both. In the first case we obtain the figures expressed in equations (4). In the second case we obtained the values in equation (7) and (8).

Pg9L270: “UCAG, from CAG1, for..” should be “UCAG, and from CAG1, for..”

Response: we apologize for the misprint, the sentence has been corrected.

Pg9L270/ Table 5: How are these standard deviations computed? Is it based on the epoch-by-epoch solution from RTKLib? Convert these stds to ENU.

Response: Standard deviations come from estimates of the daily solutions. We have added an explicative sentence from line 253 to line 258. As requested, we converted to ENU and added them in Table 5.

In the processing of UFF1 and POEM, did you apply the antenna phase centre offsets for the Trimble and ASH701945C_M antennas? How did you verify that the antenna offsets are applied correctly?

Response: In processing both stations, we applied the corrections to the phase centers contained in the igs14.atx file. This information has been better specified in lines from 252 to 253. We know that the correction was made because the two stations (POEM and UFF1) were calculated independently by both CAG1 and UCAG. If the correction had not been made the loop closure error from the two determinations would have been much greater than what we had. In fact, CAG1 has a phase center offset of as much as 219.89 mm and UCAG has a phase center offset of 164.15 mm.

Section 3.3 is missing – does it start at Pg10 L299?

Pg10 L299-300: This sentence makes no sense – rewrite.

Response: We apologize for the typo, the error has been corrected and the first sentence of paragraph 3.3 has been corrected as well.

Pg10 L306 onwards: I am not convinced the method of assessing the RT position accuracy of the car tests gives a true reflection of the accuracy.

Response: Again, we completely agree with the referee, as written in lines 459-462, but we reinforced the thought with a further comment.

Reviewer 2 Report

Dear Authors,

The experimentation is of high interest for GNSS users. Following points may be considered:

  1. Thorough English check is required.
  2. Abstract: ln 21- replace “where” with “were”…
  3. Abstract: “good candidate” shall be replaced with some statistical results of the study.
  4. “The presented work/study”… can be used instead of”… our work…”.
  5. Line 104: “two new reference stations were established for….” Will be better than “…We set up two new reference stations and made two car tests….”
  6. Variables in all the formulas shall be explained clearly: (????+)++0
  7. The processing of Reference stations established by authors /Italian agency may be detailed to the extent possible. Does it use IGS data also for processing?
  8. Flowchart shall be added in the Method section.
  9. Results and Discussion / conclusion sections can be enhanced accordingly (as per above points/changes carried out).

Thanks.

best wishes,

Author Response

The experimentation is of high interest for GNSS users. Following points may be considered:

Response: We thank the reviewer for the attentive reading and the positive comments. We have tried our best to address all the reviewer’s comments and significantly improve the paper.

  1. Thorough English check is required.
  2. Abstract: ln 21- replace “where” with “were”…
  3. Abstract: “good candidate” shall be replaced with some statistical results of the study.
  4. “The presented work/study”… can be used instead of”… our work…”.
  5. Line 104: “two new reference stations were established for….” Will be better than “…We set up two new reference stations and made two car tests….”

Response: All suggestions were taken on board, and the entire text was revised by a native speaker.

Variables in all the formulas shall be explained clearly: (????+)++0

Response: The authors thank you for the suggestion. The paragraph has been rewritten. Formulas (7) now refer to real-time values, while formulas (8) refer to those obtained in post-processing (lines from 289 to 293).

The processing of Reference stations established by authors /Italian agency may be detailed to the extent possible. Does it use IGS data also for processing?

Response: thanks for pointing out the oversight, yes we have used the igs14.atx file for receiver geodetic antennas PCV correction. A row with this information has been added . Moreover the coordinate assessment of the reference stations has been expanded.

Flowchart shall be added in the Method section.

Response: a flow chart with the corresponding text has been added (lines from 112 to 127)

Results and Discussion / conclusion sections can be enhanced accordingly (as per above points/changes carried out).

Response: Results and discussion were added in multiple lines

Round 2

Reviewer 1 Report

I still feel there is not enough novelty in the paper to deserve publication. The method of assessing RTK positioning (comparing real-time vs post processed) is still not sufficient in my opinion.

Author Response

We wish to express our deep gratitude to the reviewers and the editor for their time to read the manuscript. The manuscript has been revised based on the reviewers’ comments. The corresponding changes are made in the attached manuscript and marked in green.

I still feel there is not enough novelty in the paper to deserve publication. The method of assessing RTK positioning (comparing real-time vs post processed) is still not sufficient in my opinion.

Response: We would like to thank the reviewer for his continued effort in helping us to improve our paper. The work carried out in this part of the research has mainly the purpose of testing the use of the u-blox receiver to upgrade a research (measurement of tropospheric delays) and professional GPS network infrastructure.

The comparison between the RTK solutions obtained in real time and the post-processed ones took place following an experiment in which we performed real-time positioning on points of up to 30 km away and for extended periods of time. In an attempt to further clarify the usefulness of our research, we haveintroduced elements of further comparison between the results obtained in real time and those reprocessed with RTKLib. The considerations arising from the comparison have been further specified in the conclusions.

Reviewer 2 Report

Dear Authors,

Manuscript has been improved however, still improvements are needed:

  1. Abstract: Sentence should be checked for the intended meaning: "...less then 5% of differences were below 30 mm in the horizontal component and 100 mm in the vertical component....".
  2. Possibly Result section can be combined with Discussion (as Results and Discussion section) for clear inferences with all figures/graphs.
  3. Conclusion shall be added in the manuscript.
  4. Latest relevant References can also be added.

best wishes,

Author Response

We wish to express our deep gratitude to the reviewers and the editor for their time to read the manuscript. The manuscript has been revised based on the reviewers’ comments. The corresponding changes are made in the attached manuscript and marked in green.

  1. Abstract: Sentence should be checked for the intended meaning: "...less then 5% of differences were below 30 mm in the horizontal component and 100 mm in the vertical component....".

Response: We apologize for the oversight; the meaning of the sentence was the opposite to what we intended. The error has been corrected.

  1. Possibly Result section can be combined with Discussion (as Results and Discussion section) for clear inferences with all figures/graphs.

Response: Following the reviewer's advice, we reorganized section 3. The section is divided into two sub-sections. The first brings together the results of the experimentation on both stations (POEM and UFF1) and contains, rearranged, the comments on the results previously contained in the discussion. The second section brings together the results of the two car tests and the discussion on the rover experiment.

  1. Conclusion shall be added in the manuscript. Latest relevant References can also be added.

Response: Some parts of the discussion found a more proper place in the conclusions, which were enriched by some new references. We added a reference to a very recent paper (Krietemeyer, A.; van der Marel, H.; van de Giesen, N.; ten Veldhuis, M.-C. A Field Calibration Solution to Achieve High-Grade-Level Performance for Low-Cost Dual-Frequency GNSS Receivers and Antennas. Sensors 2022, 22, 2267. https://doi.org/10.3390/s22062267) which describes very well the problem also reported by us.

Round 3

Reviewer 2 Report

Dear Authors,

You have carried out the corrections now. 

The summary of the process in Conclusion makes the Conclusion bit long. No need to repeat the methodology in Conclusion. Keep the Conclusion as per the points concluded in the study. 

Separation of Material and Method section into two dedicated sections: one on 'Material' and another on 'Methodology' will make the manuscript better by proper focusing...

Few more relevant references can also be added.

Thanks.

best wishes,

Author Response

We wish to express our deep gratitude to the reviewer and the editor for their time to read the manuscript. The manuscript has been revised based on the reviewers’ comments.

Following the reviewer's advice, we reorganized sections. Some parts of the conclusions found place in materials. Some new references were added.